# Risk Assessment in Urban Large-Scale Public Spaces Using Dempster-Shafer Theory: An Empirical Study in Ningbo, China

**DOI:** 10.3390/ijerph16162942

**Published:** 2019-08-16

**Authors:** Jibiao Zhou, Xinhua Mao, Yiting Wang, Minjie Zhang, Sheng Dong

**Affiliations:** 1School of Civil and Transportation Engineering, Ningbo University of Technology, Ningbo 315211, China; 2College of Transportation Engineering, Tongji University, Shanghai 200082, China; 3Intelligent Transport System (ITS) R & D Center, Shanghai Urban Construction Design and Research Institute (Group) Co., Ltd., Shanghai 200082, China; 4School of Economics and Management, Chang’an University, Xi’an 710064, China; 5Department of Civil and Environmental Engineering, University of Waterloo, Waterloo, ON N2L 3G1, Canada

**Keywords:** traffic safety, risk assessment, Dempster-Shafer theory, normal cloud model, index system, large-scale public spaces

## Abstract

Urban Large-scale Public Spaces (ULPS) are important areas of urban culture and economic development, which are also places of the potential safety hazard. ULPS safety assessment has played a crucial role in the theory and practice of urban sustainable development. The primary objective of this study is to explore the interaction between ULPS safety risk and its influencing factors. In the first stage, an index sensitivity analysis method was applied to calculate and identify the safety risk assessment index system. Next, a Delphi method and information entropy method were also applied to collect and calculate the weight of risk assessment indicators. In the second stage, a Dempster-Shafer Theory (DST) method with evidence fusion technique was utilized to analyze the interaction between the ULPS safety risk level and the multiple-index variables, measured by four observed performance indicators, i.e., environmental factor, human factor, equipment factor, and management factor. Finally, an empirical study of DST approach for ULPS safety performance analysis was presented.

## 1. Introduction

Urban Large-scale Public Spaces (ULPS) are important carriers of urban culture and economic development, which are also an indispensable part of the cultural and economic life of the citizens. With the continuous increase of urban population and the higher frequency of social interaction, a variety of large-scale activities have promoted the construction of ULPS (e.g., theaters, stadiums, railway stations, subway stations, commercial streets, and supermarkets, etc.). According to the rough statistics in the ULPS all over the world from 2000 to 2018 shown in Table 1 [1], many kinds of risk incidents such as fire accidents and trampling incidents often occur. Therefore, the risk assessment issue in large-scale public spaces has become a hot topic at present [2].

Urban large-scale public spaces safety assessment has played a crucial role in the theory and practice of urban sustainable development. It is necessary to identify potential safety hazards in large-scale public spaces, so that relevant management departments would take measures to avoid risks promptly. It is of great significance to carry out research on safety risk assessment in ULPS, as follows: (1) to evaluate the current situation of ULPS and promote the construction of safety guarantee system; (2) to prevent the occurrence of various risk events and carry out active safety control; and (3) to promote the spread of risk knowledge in ULPS, and raise public awareness of risks. In addition, it also has important theoretical value and social significance for promoting the development of urban disaster prevention and mitigation.

Due to ULPS’s own architectural structure, spatial layout, personnel and wealth distribution, ULPS are vulnerable to various internal and external risks, such as fire, flood, explosion, and crowding. A large number of facts [1,2] have proved that, although the ULPS have a low probability of occurrence, it will inevitably cause irreparable casualties or property losses in the event of accidents. Hence, it is very helpful to identify and evaluate the possible risks in ULPS, which will not only help to assess the current situation of large-scale public places and promote the construction of safety guarantee system, but also prevent the occurrence of various risky incidents and carry out active safety regulation, and improve the level of information about risk knowledge of large-scale public spaces and enhance public awareness of risks.

Unfortunately, at present, although urban managers attach great importance to the safety risk assessment of ULPS, there are many shortcomings in the current research, such as risk assessment index system, advanced and feasible assessment methods. On the one hand, ULPS are important areas of urban culture and economic development, which are characterized by high-density gathering, high mobility, and high concentration, potential safety risks are associated with operations and production activities in large public spaces. On the other hand, the safety risk assessment of ULPS is a complex process involving multiple factors, multiple indicators, and uncertainties, many indicators are often difficult to accurately quantify and compare the risk analysis.

To fill this gap, we have divided the whole evacuation process for a ULPS into two periods. For the first period, we proposed a two-level risk assessment index system for the ULPS, which is calculated and identified by an index sensitivity analysis method; and next, a Delphi method and information entropy method were also applied to collect and calculate the weight of risk assessment indicators. For the second period, we have employed a Dempster-Shafer Theory (DST) method to integrate multi-source uncertainty information and apply this method to the security risk assessment of large-scale public spaces, which also helps calculate joint information from the sets of mass sources and conduct evidence fusion for all levels of risk indicators.

This research makes the following two contributions. Firstly, a scientific and practical risk assessment indicator system (RAIS) for ULPS was proposed, which includes four first-level indicators and 20 second-level indicators. Secondly, a Dempster-Shafer Theory (DST) with evidence fusion technique was employed to analyze the interaction between the RAIS and risk level in the ULPS. Conclusively, an empirical study of DST approach for ULPS safety performance analysis was verified and presented.

The remainder of this paper is organized as follows. Section 2 reviews the existing studies on risk analysis method and visualization of risk analysis. Section 3 introduces the Dempster-Shafer theory, identifies the risk assessment index system and determines the weight of risk assessment indicators. Section 4 describes a numerical example and presents the risk assessment results. Section 5 discusses the results obtained from the model. Section 6 draws the conclusions.

## 2. Related Studies

### 2.1. Risk and Risk Assessment Analysis

Risk analysis is related to the survival of humans and loss of property on ULPS, and during public place risk assessment, “risk” is associated with a number of factors or indicators [3], such as environmental factors, human factors, infrastructure factors, management factors and known as a public place safety outcome. In the field of urban large-scale public spaces safety, the “risk assessment” [4] is defined as identify and prevent any risk(s) associated with a decision and evaluating all possible outcomes and potential impacts of risk. In general, there are three steps for risk assessment analysis [5]. First, the risk influencing factors are analyzed; second, a reasonable and scientific risk assessment index system is established; third, a scientific risk assessment method is selected to form a comprehensive evaluation process based on the above two steps. It is the capacity to see problems as they arise, deal with them and try to prevent them from happening again.

There are a number of different risk analysis methods for researchers to choose from [6,7,8]. At present, they can be divided into three categories, i.e., qualitative risk assessment methods, semi-quantitative risk assessment methods, and quantitative risk assessment methods. For the qualitative risk assessment methods [9,10,11], which are easy to operate, the evaluation process and results are simple to express, such as (a) questionnaire survey method; (b) collective discussion method; (c) expert investigation method; (d) safety checklist method; and (e) risk assessment matrix (RAM). However, these methods rely more on the experience of the evaluators and lack depth in describing the risk of the system. The evaluation results of different types of evaluation objects are not comparable. Meanwhile, they have strict requirements on the professionalism of the personnel involved in the risk assessment.

For the semi-quantitative risk assessment methods, such as (a) fault tree analysis [12,13,14]; (b) event tree analysis [15]; (c) preliminary hazard analysis [16,17]; (d) failure mode and effect analysis [18,19]; and (e) index evaluation method [20,21], they are easy to implement and have stronger objectivity than qualitative analysis, but need a lot of preparatory work. For example, these methods can be used to evaluate the probability of accidents and risk analysis, but preparatory work such as (a) estimating the exposure time of people in a dangerous environment, (b) evaluating the severity of accidents, (c) determining the corresponding scores of different factors, and (d) drawing the evaluation conclusions, are needed to be prepared in advance.

As for the quantitative risk assessment methods, they can reflect the risk levels by mathematical calculation, such as (a) fuzzy comprehensive evaluation method [13]; (b) factor analysis [22,23]; (c) analytic hierarchy process [24]; (d) cluster analysis [25,26]; (e) regression analysis [27,28,29]; (f) Bayesian analysis [30,31,32]; (g) logit model [33,34,35]; (h) time series analysis method [36,37]; and (i) Dempster-Shafer theory [38,39]. These methods have an objective and direct evaluation through the observed parameters, so as to find out the dangerous and harmful factors in the evaluation system, and then put forward the corresponding solutions in technology and management to realize safety management. 

### 2.2. DST and Applications

Dempster-Shafer theory (DST) is a general framework for reasoning with uncertainty by Dempster [38] in 1967, and the theory was later developed by Shafer [39] into a general framework for modelling epistemic uncertainty. Previously, risk assessment of ULPS has many characteristics such as multiple indicators and multiple knowledge areas, meanwhile, the risk assessment process has a lot of uncertainties and is difficult to strictly quantify. Fortunately, DST provides an opportunity to combine multiple evidence sources. This is because DST is an established technique that maintains a mechanism of multiple indicators inputs and integrated outputs for the evaluation of risk analysis.

DST is a technique that can handle all the available evidence from different sources and arrive at a degree of belief (represented by a mathematical object called belief function) [40,41,42,43]. Previously, DST is mainly used in artificial intelligence, pattern recognition, data fusion technology, expert system and decision analysis, and its application in risk assessment has gradually increased in recent years. For example, Sun et al. developed an alternative methodology for the risk analysis of information systems security under the DST of belief functions. Zeng et al. [44] proposed a technique for traffic incident detection, which combined multiple multi-class probability support vector machines using DST approach. Rassafi et al. [45] employed an DST approach for road safety assessment modelling as a complex multi attribute decision analysis problem to deal with unavoidable uncertainties such as ignorance and vagueness. Through the application of DST, we can analyse the multiple-index variables related to safety situations in urban large-scale public spaces. Analysis of risk level with reference to risk assessment indicators will be helpful in understanding safety performance of ULPSs. 

### 2.3. Visualization of Risk Analysis Studies

From the web of science database, 1997–2019, we have searched the main research countries, authors, and research institutions in risk analysis studies, our search topics are “traffic safety” and “risk analysis”, and we found that there were a total of 1400 + records, the number of authors was 4515, the number of research institutions was 1488, and the number of countries was 91. Next, the visualization of keywords in risk analysis studies, 1997–2019, were created by VOSviewer software [46], as shown in Figure 1. 

The keywords are an important part of a research paper, which carries important information about the fields of interests. A total of 5000+ keywords appeared in all the collected literature on traffic safety and risk analysis, see Figure 1. Figure 1a is the density of main research keywords and Figure 1b is the keywords co-occurrence network of risk analysis studies. It can be clearly seen that the research theme of risk assessment has roughly formed four clusters in Figure 1b, and there is a significant correlation between the keywords in each cluster.

The visualization of keywords in risk analysis studies indicates that previous studies [8,13,16,19,20,25,26] that focused on risk assessment in ULPS were undertaken at an aggregate level. Systematization is the key requirement in safety risk assessment. However, systematization firstly means the evaluation system is systematic, which reflects the performance of specific models; secondly, the evaluation system should comprehensively cover both qualitative and quantitative indicators. Most ULPS safety risk studies [8,13,16,19,20,25,26] rely on questionnaire survey data but do not provide much information about the index system of risk assessment. In addition, little is known about the underlying factors that affect the risk analysis of ULPS. Furthermore, there is a paucity of research discerning the interrelationships between the multiple-index system and risk assessment, and the contributory indicators and multiple-index weight at a disaggregated level using Delphi method and entropy method.

## 3. Method

### 3.1. Dempster-Shafer Theory (DST)

#### 3.1.1. Framework of Discernment (FoD)

DST is a finite nonempty set of hypotheses as the frame of discernment (FoD) Θ. Let Θ be the universe: the set representing all possible states of a system under consideration. The power set $2^{\Theta }$ is the set of all subsets of Θ, including the empty set ϕ. For example, if Θ={a,b}, then $2^{\Theta }=\{\phi ,\{a\},\{b\},\Theta \}$.

The elements of the power set can be taken to represent propositions concerning the actual state of the system, by containing all and only the states in which the proposition is true.

#### 3.1.2. Basic Belief Assignment (BBA)

The theory of evidence assigns a belief mass to each element of the power set. Formally, a function $m:2^{\Theta }\to [0,1]$ is called a basic belief assignment (BBA), which has two properties. First, the mass of the empty set is zero. (1)m(Φ)=0

Second, the masses of the remaining members of the power set add up to a total of 1. (2)$$\sum_{A\subseteq 2^{\Theta }}m(A)=1$$

The mass *m*(*A*) of *A*, a given member of the power set, expresses the proportion of all relevant and available evidence that supports the claim that the actual state belongs to *A* but to no particular subset of *A*. The value of *m*(*A*) pertains only to the set *A* and makes no additional claims about any subsets of *A*, each of which have, by definition, their own mass.

From the mass assignments, the upper and lower bounds of a probability interval can be defined. This interval contains the precise probability of a set of interest (in the classical sense), and is bounded by two non-additive continuous measures called belief (or support) and plausibility.

The belief bel(*A*) for a set *A* is defined as the sum of all the masses of subsets of the set of interest. (3)$$bel(A)=\sum_{B|B\subseteq A}m(B)$$

The plausibility pl(A) is the sum of all the masses of the sets B that intersect the set of interest A. (4)$$pl(A)=\sum_{B|B\cap A\neq \emptyset }m(B)$$

The two measures are related to each other as follows. (5)$$pl(A)=1-bel(\bar{A})$$

And conversely, for finite *A*, given the belief measure *bel*(*B*) for all subsets *B* of *A*, we can find the masses *m*(*A*) with the following inverse function. (6)$$m(A)=\sum_{B|B\subseteq A}{\left(-1\right)}^{\left|A-B\right|}bel(B)$$where |A−B| is the difference of the cardinalities of the two sets.

#### 3.1.3. Dempster’s Combination Rule

Generally, the detailed combination rule is to calculate the distance between the two sets of masses *m_i_* and *m_j_*, and then modify the obtained mass function by the calibration coefficient. Finally, the combination rule is used for evidence fusion in the following manner.

Step1: The Θ is a finite nonempty set of hypotheses as the FoD, the masses mi and mj are the base degrees of belief (or confidence, or trust) for the frame of discernment Θ, the distance between the two sets of masses *m_i_* and *m_j_* is calculated by Equation (7). (7)$$d_{ij}=\sqrt{(1/2)\times {\left(m_{i}-m_{j}\right)}^{T}\times D(m_{i}-m_{j})}$$ where *D* is a matrix of 2*^N^*×2*^N^*, the element in the matrix is, *D*(*A_i_*, *A_j_*)=(|*A_i_* ∩ *A_j_*|)/(|*A_i_* ∪ *A_j_*|), *i* = 1, 2, … , 2*^N^*, and *d_ij_* indicates the difference between the two sets of masses, *d_ij_*∈[0,1].

The similarity between the sets of masses *m_i_* and *m_j_* is *S_ij_*, as shown in Equation (8). (8)$$S_{ij}=1-d_{ij}$$

The MASS function $M_{mi}(R_{j})$ is compared in pairs, and the distance between the evidences is calculated to obtain the evidence similarity matrix *Sim*. (9)$$Sim=\left[\begin{array}{cccc} 1 & s_{12} & \cdots & s_{1n}\\ s_{21} & 1 & \cdots & s_{2n}\\ \vdots & \vdots & \vdots & \vdots \\ s_{n1} & s_{n2} & \cdots & 1 \end{array}\right]$$

The degree to which evidence $m_{i}$ is supported by other evidence is described as Sup($m_{i}$). (10)$$Sup(m_{i})=\sum_{j=1,j\neq i}^{n}S_{ij}$$

The credibility of the evidence $m_{i}$ is $Crd_{1}(m_{i})$. (11)$$Crd_{1}(m_{i})=\frac{Sup(m_{i})}{max\{\underset{1\leq k\leq n}{Sup(m_{k})}\}}$$

Due to the large number of professional fields covered by safety risk assessment in ULPS, experts in various fields have strong professional knowledge background and authority. In order to make the evaluation results more accurate, comprehensive consideration of each expert’s professional background and other factors, we give each expert the same weight [33,34] for the RAIS, the sum of weights is $\sum_{r=1}^{n}\lambda_{r}=1$. From this, the credibility of expert i is $Crd_{2}(m_{i})$. (12)$$Crd_{2}(m_{i})=\frac{Sup(\lambda_{i})}{max\{\underset{1\leq k\leq n}{Sup(\lambda_{k})}\}}$$

Step 2: According to the credibility of the evidence and the credibility of the experts, the calibration coefficient $\alpha_{i}$ of the evidence is obtained. (13)$$\alpha_{i}=\mu Crd_{1}(m_{i})+(1-\mu )Crd_{2}(m_{i})$$

Supposing μ=0.5, it means that the credibility of evidence and the credibility of experts are of equal importance.

According to the calibration of coefficient $\alpha_{i}$, we adjust the MASS function of the indicator $B_{i}$ shown in Equations (14) and (15). (14)m(B)=αM(B)
(15)m(Θ)=αM(Θ)+1−α

After modification by Equations (14) and (15), a new MASS function $m_{Bi}(R_{j})$ is obtained, then the Dempster’s rule of combination is used to merge the *m* experts’ comments on the second-level index $B_{i}$, and the new MASS function $m_{Bi}(R_{j})$ is also obtained corresponding to each second-level index $B_{i}$.

Step3: Dempster’s orthogonal rule of combination is determined by Equations (16) and (17). (16)$$m(A)=K^{-1}\sum_{A_{1}\cap A_{2}\cap \cdots \cap A_{n}=A}m_{1}(A_{1})m_{2}(A_{2})\cdots m_{n}(A_{n})$$
(17)$$K=1-\sum_{A_{1}\cap A_{2}\cap \cdots \cap A_{n}=\varphi }m_{1}(A_{1})m_{2}(A_{2})\cdots m_{n}(A_{n})\;=\sum_{A_{1}\cap A_{2}\cap \cdots \cap A_{n}\neq \varphi }m_{1}(A_{1})m_{2}(A_{2})\cdots m_{n}(A_{n})$$where *K* is a measure of the amount of conflict between the two mass sets, if *K* = 0, it means that all evidence is completely contradictory, and the Dempster’s combination rule cannot be applied; Conversely, if *K*≠0, then the fusion of a set of evidence $m_{1},m_{2},\cdots ,m_{n}$ becomes the orthogonal sum, *m* is new evidence produced by the combination, and it also is a MASS function too. Note that $m=m_{1}⊕m_{2}⊕m_{3}\cdots ⊕m_{n}$, which represents the combination of $m_{1}$, $m_{2}$, and $m_{n}$, carries the joint information from the sets of masses sources.

According to the principle of maximum membership degree, the MASS function of the second-level indicator and its weight are linearly weighted, and the risk level of the first-level indicator can be obtained, see Equation (18). (18)$$F=\sum_{i=1}^{n}\omega_{i}m_{B_{i}}(R_{j})$$

Similarly, the MASS function $M_{Ai}(R_{j})$ corresponding to each level indicator $A_{i}$ can be obtained, and the risk evaluation level of the large-scale public space is determined.

### 3.2. Risk Assessment Index System (RAIS)

#### 3.2.1. Preliminary RAIS

The safety of the urban large-scale public spaces will be affected by various factors during the operation, especially when the high-density pedestrian flow is gathering. RAIS has played a crucial role in the risk assessment of ULPS’s development, while the scientific knowledge and comprehensiveness of index system will directly affect the accuracy of safety risk assessment in the ULPS. The principles [20,21] for selecting the preliminary RAIS in ULPS are as follows, 

(1) Principle of scientific

All indicators in the RAIS can objectively reflect the risk factors faced by ULPS. The division of indicators has the basis of scientific theory for reference and can truly reflect the characteristics of security risks in ULPS.

(2) Principle of comprehensiveness

All indicators in the RAIS can comprehensively reflect the specific situation of security risks in ULPS. The selection of the preliminary risk indicators in ULPS not only includes the internal and external risk factors of the buildings, but also includes the surrounding environment, transportation facilities, system management, and other factors. Generally speaking, the selection of RAIS covers all factors affecting the security of ULPS.

(3) Principle of accuracy

All indicators in the RAIS are interrelated and independent, so as to ensure that the same type of evaluation indicators is not repeated, and indicators that have weak influence on security risks in ULPS should be eliminated to avoid statistical difficulties, calculation redundancy and credibility reduction.

(4) Principle of operability

All indicators in the RAIS need to be clear and easy to understand, and easy to conduct questionnaires or data collection. Principle of operability is to ensure the authenticity, effectiveness and operability of the risk assessment.

At the same time, based on the above principles and previous literatures [13,16,19,20,21,22,23,25,26,27,43,44,45], this study built a risk assessment index system including 4 first-level indicators, i.e., environmental factors, human factors, infrastructure factors and management factors, and 31 second-level indicators, as shown in Table 2.

#### 3.2.2. Sensitivity Analysis of Indicators

The risk assessment indicators established in the initial stage covered too much information, which not only caused the information redundancy but also increased the difficulty of risk assessment and reduced the accuracy of the evaluation results. Therefore, the indicators should be screened. 

The Delphi method was used to analyze the sensitivity of evaluation indicators. In the process of sensitivity analysis, we take the factors that have a greater impact on the risk assessment as the sensitivity index, our purpose is to find the sensitive indicators and delete the non-sensitive indicators. 

Let ${\bar{E}}_{ip}$ as the ith indicator of the degree of acceptance in the expert group p, let ${\bar{E}}_{i}$ as the ith indicator of the total average degree of acceptance in the expert group n. The formula is as follows:(19)$${\bar{E}}_{ip}=\frac{1}{k}\sum_{j=1}^{5}E_{j}n_{ijp}$$
(20)$${\bar{E}}_{i}=\frac{1}{n}\sum_{p=1}^{n}E_{ip}$$where $n_{ijp}$ is the number of experts who are considered to be j-level in the expert group p; $E_{j}$ is the corresponding value of the j-level importance of a certain indicator. The importance of indicators is divided into five levels: very unimportant (E_1_ = 1), unimportant (E_2_ = 2), generally important (E_3_ = 3), important (E_4_ = 4), very important (E_5_ = 5). Through the calculations of Equations (19) and (20), the total average acceptance of each indicator in all expert groups is calculated. The result of the sensitivity analysis is shown in Figure 2.

After removing the indicator with the value less than 4.0, the final 20 indicators are obtained, as shown in Figure 3.

### 3.3. Determination of the Weight of Risk Assessment Indicators

In the process of risk assessment, the importance of various indicators in the evaluation system is different. Hence, it is necessary to establish the corresponding weights of indicators for ULPS. The entropy weight method is used to calculate the index weights, that is, firstly, the Delphi method [20,21] is used to collect the weight information by experts; secondly, the value of the ranking matrix of expert evaluation is calculated; thirdly, the entropy value is calculated by the entropy decision process, and final weights of indicators are obtained by the entropy weight method [20,21,22,30].

#### 3.3.1. Collection of Expert Opinions for RAIS

The weight information is collected using Delphi method by experts. It is assumed that *m* experts are invited to participate in the weight information survey of the safety risk assessment indicators in LSPS, the survey requires that the experts hired to sort the evaluation index sets according to their rich knowledge, professionalism and practical experience. Five indicates that the index is "most important" and four indicates "important", and the importance decreases in turn until 1, allowing experts to have the same value for multiple indicators. The ranking matrix obtained by *m* experts is set to *R*. (21)$$R=\left(\begin{array}{cccc} a_{11} & a_{12} & \cdots & a_{1n}\\ a_{21} & a_{22} & \cdots & a_{2n}\\ \vdots & \vdots & \vdots & \vdots \\ a_{m1} & a_{m2} & \cdots & a_{mn} \end{array}\right)$$where $a_{mn}$ is the evaluation value of the *m*th expert for the *n*th indicator.

#### 3.3.2. Calculation of Indicator Weight

To determine the entropy value, the ranking matrix is first transformed into the membership matrix [20,21]. The membership function of ranking transformation is defined as S(G)
(22)$$S(G)=-\lambda x_{n}(G)Inx_{n}(G)$$where (23)$$x_{n}(G)=\frac{N-G}{N-1},\lambda =\frac{1}{In(N-1)},N=n+2$$

We define P(G) as membership function, as shown in Equation (24), (24)$$P(G)=S(G)/(\frac{N-G}{N-1})-1$$

Then, (25)$$P(G)=\frac{In(N-G)}{In(N-1)}$$where *G* is the ranking value given by the experts, *N* is the index value after standardization conversion, and *n* is the number of indicators.

The ranking number of each index is brought into the Equation (25), and the sorting matrix *M* can be converted into the membership degree matrix $M={\left(q_{ij}\right)}_{m\times n}$, and $q_{ij}$ is called the membership degree of the ranking number *G*. Taking the column vectors in the membership matrix M as $q_{j}={\left(q_{1j},q_{2j},\cdots ,q_{mj}\right)}^{T}$, and the average value $\bar{q_{j}}$ of the membership in the vector is obtained, as shown in Equation (26). (26)$$\bar{q_{j}}=\frac{1}{m}\sum_{i=1}^{m}q_{mj}$$

Therefore, the mean square deviation $S_{j}$ of the membership in the vector can be obtained. (27)$$S_{j}=\sqrt{\frac{1}{m}\sum_{i=1}^{m}{\left(q_{mj}-q_{j}\right)}^{2}}$$

The comprehensive evaluation value of each index by m experts is recorded as $\sigma_{j}$, as shown in Equation (28). (28)$$\sigma_{j}=\bar{q_{j}}(1-S_{j})$$

Thereby, the risk evaluation vector $\theta =(\sigma_{1},\sigma_{2},\cdots ,\sigma_{n})$ of index $\sigma_{j}$ can be obtained.

We normalize the weights $\omega_{j}$ of indicators, as shown in Equation (29). (29)$$\omega_{j}=\sigma_{j}/\sum_{j=1}^{n}\sigma_{j}$$

Let $W=(\omega_{1},\omega_{2},\cdots ,\omega_{n})$ be the weight vector of risk assessment index system $U=\{u_{1},u_{2},\cdots ,u_{n}\}$, $\omega_{j}>0(j=1,2,\cdots ,n)$, and $\sum_{j=1}^{n}\omega_{j}=1$. Each value in *W* corresponds to the weight of each grade of indicators, and the larger the value, the stronger the impact of the index on the safety risk assessment of large-scale public spaces.

## 4. Case Study

Tian-yi Square, the largest commercial plaza in Ningbo City, was completed by the end of 2001, the total investment of the project was 1.25 billion RMB. It has 167,000 square meters of shops, 20,000 square meters of parking lots, 64,000 square meters of green space, 6,000 square meters of water areas and 1,000 square meters of performing stage. In the case study, we take the Tian-yi Square, one of the biggest large-scale public spaces in Ningbo, as an example, the DST method is proposed to evaluate the safety risk of ULPS, in order to verify the practicability and effectiveness of the risk assessment method.

### 4.1. Data Collection

We developed an expert questionnaire based on the RAIS given above. Experts from various research fields such as traffic engineering, traffic safety, municipal engineering and risk management were invited to evaluate the risk assessment indicators. According to the collected data, scores of RAIS by experts as shown in Table 3.

### 4.2. Calculation of Index Weight for RAIS

Experts’ decision-making is an important part of index weight calculation and risk assessment for the ULPS. In order to reduce the influence of different experts on the evaluation results, firstly, the assumption is that we fully believe and respect the scoring results of the experts, secondly, we employ as many experts from different fields as possible to score together and take the average, according to their rich knowledge, professionalism and practical experience. In addition, in the actual calculation process, we remove a maximum score, while removing a minimum score, and then calculate the average of the remaining index scores. According to the scores of expert survey results (see Table 2), the ranking matrix of expert opinions can be obtained.


$M=\left(\begin{array}{cccc} 2 & 2 & 3 & 1\\ 3 & 2 & 2 & 1\\ 4 & 2 & 3 & 1\\ 2 & 3 & 1 & 1\\ 3 & 1 & 3 & 2 \end{array}\right)$
$M_{1}=\left(\begin{array}{ccccc} 5 & 2 & 3 & 1 & 4\\ 4 & 2 & 3 & 1 & 5\\ 2 & 3 & 4 & 1 & 5\\ 3 & 2 & 4 & 3 & 5\\ 5 & 4 & 3 & 2 & 4 \end{array}\right)$
$M_{2}=\left(\begin{array}{ccccc} 3 & 4 & 1 & 5 & 2\\ 3 & 1 & 2 & 5 & 4\\ 2 & 4 & 2 & 4 & 5\\ 3 & 3 & 4 & 5 & 2\\ 1 & 4 & 2 & 4 & 5 \end{array}\right)$



$M_{3}=\left(\begin{array}{ccccc} 2 & 5 & 3 & 4 & 1\\ 1 & 3 & 3 & 4 & 2\\ 1 & 2 & 4 & 4 & 2\\ 2 & 4 & 3 & 5 & 1\\ 1 & 2 & 3 & 4 & 2 \end{array}\right)$
$M_{4}=\left(\begin{array}{ccccc} 1 & 4 & 3 & 2 & 2\\ 2 & 2 & 4 & 3 & 1\\ 2 & 1 & 4 & 3 & 2\\ 1 & 2 & 4 & 3 & 1\\ 2 & 3 & 5 & 4 & 1 \end{array}\right)$


The ranking membership matrix M of each grade of indicators can be obtained by Equations (21)–(24).


$M=\left(\begin{array}{cccc} 0.861 & 0.861 & 0.683 & 1.000\\ 0.683 & 0.861 & 0.861 & 1.000\\ 0.431 & 0.861 & 0.683 & 1.000\\ 0.861 & 0.683 & 1.000 & 1.000\\ 0.683 & 1.000 & 0.683 & 0.861 \end{array}\right)$



$M_{1}=\left(\begin{array}{ccccc} 0.387 & 0.898 & 0.774 & 1.000 & 0.613\\ 0.613 & 0.898 & 0.774 & 1.000 & 0.387\\ 0.898 & 0.774 & 0.613 & 1.000 & 0.387\\ 0.774 & 0.898 & 0.613 & 0.774 & 0.387\\ 0.387 & 0.613 & 0.774 & 0.898 & 0.613 \end{array}\right)$
$M_{2}=\left(\begin{array}{ccccc} 0.774 & 0.613 & 1.000 & 0.387 & 0.898\\ 0.774 & 1.000 & 0.898 & 0.387 & 0.613\\ 0.898 & 0.613 & 0.898 & 0.613 & 0.387\\ 0.774 & 0.774 & 0.613 & 0.387 & 0.898\\ 1.000 & 0.613 & 0.898 & 0.613 & 0.387 \end{array}\right)$



$M_{3}=\left(\begin{array}{ccccc} 0.898 & 0.387 & 0.774 & 0.613 & 1.000\\ 1.000 & 0.774 & 0.774 & 0.613 & 0.898\\ 1.000 & 0.898 & 0.613 & 0.613 & 0.898\\ 0.898 & 0.613 & 0.774 & 0.387 & 1.000\\ 1.000 & 0.898 & 0.774 & 0.613 & 0.898 \end{array}\right)$
$M_{4}=\left(\begin{array}{ccccc} 1.000 & 0.613 & 0.774 & 0.898 & 0.898\\ 0.898 & 0.898 & 0.613 & 0.774 & 1.000\\ 0.898 & 1.000 & 0.613 & 0.774 & 0.898\\ 1.000 & 0.898 & 0.613 & 0.774 & 1.000\\ 0.898 & 0.774 & 0.387 & 0.613 & 1.000 \end{array}\right)$


The mean square deviation $S_{j}$ of the membership matrix *M* can be obtained by Equations (25) and (26).

S=(0.158,0.101,0.129,0.055); $S_{1}=(0.205,0.112,0.079,0.089,0.111)$; 

$S_{2}=(0.092,0.152,0.130,0.111,0.229)$; $S_{3}=(0.050,0.194,0.064,0.191,0.050)$


$S_{4}=(0.050,0.133,0.123,0.091,0.050)$


Then, the evaluation vector $\sigma_{i}$ of each grade of indicators is obtained by Equation (27).

σ=(0.592,0.767,0.618,0.918); $\sigma_{1}=(0.487,0.725,0.654,0.851,0.424)$

$\sigma_{2}=(0.766,0.613,0.749,0.424,0.491)$; $\sigma_{3}=(0.911,0.575,0.694,0.516,0.892)$


$\sigma_{4}=(0.892,0.726,0.526,0.697,0.911)$


Finally, the weight vector $\omega_{i}$ of each grade indicator is obtained by Equation (28).

ω=(0.200,0.259,0.230,0.310); $\omega_{1}=(0.155,0.231,0.208,0.271,0.135)$

$\omega_{2}=(0.252,0.201,0.246,0.139,0.161)$; $\omega_{3}=(0.254,0.160,0.193,0.144,0.249)$


$\omega_{4}=(0.238,0.193,0.140,0.186,0.243)$


Results of the weights of primary indicators and secondary indicators are calculated by Equations (21)–(28), as shown in Table 4.

### 4.3. Evidence fusion Process of Security Risk Assessment

The membership matrix $R_{n\times j}$ of security risk level can be obtained by forward generator in Normal Cloud (NC) Model [49,50] from Table 2 and Table 3. In this study, the NC model was used to evaluation the membership matrix of security risk level. The steps to construct a NC model were as follows.

**Setp 1****:** NC model: Let U be a quantitative domain and C a qualitative concept on U. If a certain value x∈U, then x is a random implementation of C. The determinacy of x to C is a random number with a stable tendency μ(x):U→[0,1], ∀ x∈U, x→μ(x). Then, the distribution of x on the domain U is called a cloud, and each x is called a cloud droplet. Three parameters are used to characterize the cloud model, which are the expectation *Ex*, the entropy *En*, and the super-entropy He.

**Setp 2:** NC generator: If the definition of *x* in the cloud satisfies x ~ *N*(*Ex*, *En’*) and *En’* ~ *N*(*En*, *He^2^*), the determinacy of *x* to C satisfies the following. (30)$$\mu \left(x\right)=Exp\frac{-{\left(x-Ex\right)}^{2}}{2{\left(En^{\prime }\right)}^{2}}$$

The NC model is widely used in solving probabilistic and ambiguous problems, which combines the normal distribution function with the bell-shaped membership function. All the cloud models used in this study referred to the NC model. By definition, the quantitative data obtained from cloud digital eigenvalues were forward NC generators, while those obtained from the quantitative data were backward NC generators. In this study, the forward NC generator was used to solve the problem of risk level.

**Setp 3:** Cloud synthesis: The synthesis of clouds was to combine the clouds of the same nature with a parent cloud. The parent cloud C (*Ex, En, He*) was synthesized using n child clouds Cn (*Ex_n_, En_n_, He_n_*), expressed as follows. (31)$$C=C_{1}\circ C_{2}\circ C_{3}\circ \cdots \circ C_{n}$$
(32)$$Ex=\frac{\sum_{i=1}^{n}Ex_{i}\times En_{i}\times w_{i}}{\sum_{i=1}^{n}En_{i}\times w_{i}},En=\sum_{i=1}^{n}En_{i}\times w_{i},He=\frac{\sum_{i=1}^{n}He_{i}\times En_{i}\times w_{i}}{\sum_{i=1}^{n}En_{i}\times w_{i}}$$ where “◦” refers to the process of cloud synthesis and *w_i_* to the weight of the *i_th_* child cloud.

The process of the NC model followed three steps:

**Step 1**: Determining the digital features of the cloud

Let X = (*x*_1_, *x*_2_, …, $x_{j}$, …, *x*_n-1_) be the threshold vector of an index in which $x_{j}$ denotes the threshold of the index at *jth* R degree. As the synthetic clouds were conducted in the same domain, the index needed to be standardized before calculating the digital features. Using the bigger-is-better and the smaller-is-better rules, the standardizations were as shown in Equations (33) and (34), respectively.

Bigger is better, (33)$$x_{j}^{*}=\frac{x_{j}-\min\{x_{j}\}}{\max\{x_{j}\}-\min\{x_{j}\}}(j=1,2,\ldots ,n-1)$$

Smaller is better,
(34)$$x_{j}^{*}=\frac{\max\{x_{j}\}-x_{j}}{\max\{x_{j}\}-\min\{x_{j}\}}(j=1,2,\ldots ,n-1)$$ where $x_{j}^{*}$ is the standardized value of $x_{j}^{}$ , and max {$x_{j}^{}$} and min {$x_{j}^{}$} are the maximum and minimum values for the threshold j, respectively. 

The R degree *j* (*j =* 1, 2, … , *n*-1) is indicated by NCs in the semi-rising and semi-descending states, leading to the three digital features in Equation (35). Then, the other grades are indicated by NCs in full states (see Equation (36)), and their corresponding digital features are calculated as follows. (35)$$\begin{array}{l} Ex_{x1}=x_{1}^{*},En_{x1}=En_{x2}\\ Ex_{xn}=x_{n-1}^{*},En_{xn}=En_{xn-1} \end{array}$$
(36)$$\left\{ \begin{array}{l} Ex_{xj}=(x_{j-1}^{*}+x_{j}^{*})/2\\ En_{xj}=(x_{j-1}^{*}-x_{j}^{*})/6\\ He=0.01\;(j=2,\ldots ,n-1) \end{array}\right.$$

**Step 2:** Establishing the template cloud model

Synthesizing all the indexes of child clouds under the *R* degree criterion in n grade results in the parent cloud, which is called the template cloud or standardized cloud and uses a standard cloud map to assess the crowd state in the metro station. For example, a certain facility has three indexes, and each of their child cloud is denoted as *R_j_*, *S_j_*, and *T_j_*. Thus, the parent cloud *U_j_* was synthesized as *U_j_ = R_j_ ◦ S_j_ ◦ T_j_* (*j =* 1, 2, *…, n*), and can be referred from Equation (32).

**Step 3:** Establishing the candidate cloud model

(a). Establish a forward cloud generator CG*_Xj_*, according to the digital features of the R degree.

(b). Standardize the actual values *x* collected by the index and marked as *x**. Let the standardization results of index in *x* ≤ min {$x_{j}$} be 0 and index in x ≥ min {$x_{j}$} be 1.

(c) Input the standardized values *x** into the forward NC generators, please find detailed information from Refs. [49,50].

(d) Output of the cloud generator, $\mu_{Xj}$ (*j =* 1, 2, *…, n*), represents the degree of *x* belonging to *Xj*. The fuzziness and randomness nature of the NC model indicates that, instead of a determined number, $\mu_{Xj}$ is a random number with a stable tendency. The membership matrix $R_{n\times j}$ of security risk level is consisted of these $\mu_{Xj}$ for each $X_{j}$.

After normalizing all $\mu_{Xj}$, Equation (37) shows the weights $W_{Xj}$ for each $X_{j}$. (37)$$W_{Xj}=\frac{\mu_{Xj}}{\sum_{j=1}^{n}\mu_{Xj}}$$

The MASS function $M_{mi}(R_{j})$ of various indicators are obtained with these weights $W_{Xj}$. Take the secondary indicator “Architectural layout (B2)” factor as an example, as follows, the others are listed in Appendix A. $$R_{B1}=\left(\begin{array}{ccccc} 1.000 & 0.020 & 0.000 & 0.000 & 0.000\\ 0.000 & 0.020 & 0.018 & 0.000 & 0.000\\ 0.017 & 0.995 & 0.000 & 0.000 & 0.000\\ 1.000 & 0.020 & 0.000 & 0.000 & 0.000\\ 0.017 & 0.995 & 0.000 & 0.000 & 0.000 \end{array}\right)$$

Then, after normalization, the MASS function $M_{mi}(R_{j})$ of various indicators are obtained by normalized processing technology. $$M_{B1}=\left(\begin{array}{ccccc} 0.980 & 0.020 & 0.000 & 0.000 & 0.000\\ 0.000 & 0.522 & 0.478 & 0.000 & 0.000\\ 0.145 & 0.855 & 0.000 & 0.000 & 0.000\\ 0.980 & 0.020 & 0.000 & 0.000 & 0.000\\ 0.145 & 0.855 & 0.000 & 0.000 & 0.000 \end{array}\right)$$

When obtaining expert evidence comments, due to the large differences in the research fields and work experience of experts, there may be great conflicts between their evidence comments. 

Therefore, in order to solve the above problems and make the results of the evidence fusion more accurate, we use the distance function as in Equation (7) to measure the degree of conflict between pieces of evidence, and the calibration coefficients are used to revise the evidence comments from experts.

Next, the evidence similarity matrix Sim is obtained by Equations (7)–Equation (9). $$Sim=\left(\begin{array}{ccccc} 1.000 & 0.151 & 0.036 & 1.000 & 0.036\\ 0.151 & 1.000 & 0.530 & 0.151 & 0.530\\ 0.036 & 0.530 & 1.000 & 0.036 & 1.000\\ 1.000 & 1.000 & 0.036 & 1.000 & 0.036\\ 0.036 & 0.530 & 1.000 & 0.036 & 1.000 \end{array}\right)$$

And, the credibility vector $Crd_{1}$ of each expert evidence is obtained by Equations (10)∓(11). $$Crd_{1}=(0.763,0.850,1.000,0.763,1.000)$$

The credibility vector $Crd_{2}$ of experts are then obtained by expert weights
$$Crd_{2}=(1.000,1.000,1.000,1.000,1.000)$$

The calibration coefficient of the evidence is obtained by Equation (13), and the MASS function of the indicators is revised by Equation (14), the revised MASS function $m_{Bi}(R_{j})$ is obtained. $$m_{B1}=\left(\begin{array}{cccccc} 0.865 & 0.017 & 0.000 & 0.000 & 0.000 & 0.118\\ 0.000 & 0.483 & 0.442 & 0.000 & 0.000 & 0.075\\ 0.016 & 0.984 & 0.000 & 0.000 & 0.000 & 0.000\\ 0.865 & 0.017 & 0.000 & 0.000 & 0.000 & 0.119\\ 0.016 & 0.984 & 0.000 & 0.000 & 0.000 & 0.000 \end{array}\right)$$

Finally, the experts’ MASS function is merged by the D-S evidence fusion theory given in Equations (10)–(14), and the result is $m_{B_{1}}$.

The secondary indicator MASS function is obtained by Equations (7)–(14), as shown in Table 5.

According to the weight of secondary indicator and the MASS function, which are linearly weighted by the Equation (17). The result of risk level value is, 


$F_{A1}=(0.845,0.155,0.000,0.000,0.000,0.000)$
$F_{A2}=(0.861,0.139,0.000,0.000,0.000,0.000)$



$F_{A3}=(0.503,0.143,0.354,0.000,0.000,0.000)$
$F_{A4}=(0.194,0.678,0.128,0.000,0.000,0.000)$


By Equations (7)–(14), the results of the MASS function of the primary indicators is obtained by D-S evidence fusion, as shown in Table 6.

According to the weight of primary indicator and the MASS function, which are linearly weighted by the Equation (17). The result of risk level value of Tian-yi Square is, F=(0.459,0.311,0.230,0.000,0.000,0.000).

### 4.4. Outcome of the Risk Assessment

(1) The security risk level for the primary indicators can be obtained by the principle of maximum membership, as shown in Table 7.

In Table 7, for the environmental factors, the value of the excellent level is 0.845, and the value of the good level is 0.155. Overall, the risk assessment level of environmental factors is excellent. For the human factors, the value of the excellent level is 0.861, and the value of the good level is 0.139. Overall, the risk assessment level of human factors is excellent. For the infrastructure factors, the value of the excellent level is 0.503, and the value of the good level is 0.143. Overall, the risk assessment level of infrastructure factors is also excellent. For the management factor, the value of the excellent level is 0.194, and the value of the good level is 0.678. Overall, the risk assessment level of management factors is good.

(2) The overall security risk level result of Tian-yi Square is obtained by the principle of maximum membership, that is, the value of the excellent level is 0.459, the value of the good level is 0.311, and the value of the satisfactory level is 0.230. Therefore, the risk assessment level of Tian-yi Square is safety. The reason is that, Ningbo city, one of the three economic centers in Zhejiang province, is located in the southeast coastal area and belongs to the subtropical monsoon climate, at the same time, Tian-yi Square is one of the famous public places in Ningbo city, the surrounding environment is comfortable, the transportation is convenient, and society is harmonious. Hence, the overall safety risk assessment is excellent.

However, due to the early completion of Tian-yi Square, many facilities and equipment cannot be updated in time, and there are some defects in large-scale event management, store management, and staff training, so the evaluation level of infrastructure factors and management factors is only good, not excellent. 

## 5. Discussions

(1) Enhance the relevance of RAIS

The original of the risk assessment indicator system (RAIS) fully considers the factors affecting the environment, human, infrastructure and management. However, in order to improve the pertinence and applicability of the RAIS, considering the gaps in risk factors faced by different cities and different public places, the sensitivity analysis method is used to measure the entire primary RAIS in the process of index preparation. Finally, all the indicators involved in risk assessment are valuable and applicable.

The risk indicators covered in the established primary RAIS are diversified and multi-faceted, but in the end, the indicators used in the risk assessment questionnaire are screened and representative. Therefore, it is very important to improve the accuracy of risk assessment indicator system and reduce the impact of the low sensitivity of indicators to the overall risk analysis.

(2) Universality of risk assessment methods

The Dempster-Shafer Theory is used to analyze risk assessment in urban large-scale public spaces, which can effectively integrate multiple evidence. Regardless of how many secondary indicators are set in the entire risk assessment indicator system or how many experts are employed to participate in the risk assessment, the Dempster-Shafer theory can be used to combine the expert opinions and the final risk assessment results for urban large-scale public spaces is obtained. Hence, the Dempster-Shafer theory for the risk assessment has universality.

(3) Division of evaluation level

The Delphi method was used by the experts to evaluate the risk and divide the risk assessment criteria into five levels according to the principle of equidistance: excellent (8 ~ 10), good (6 ~ 8), satisfactory (4 ~ 6), fair (2 ~ 4), and poor (0 ~ 2). It is different from the two-level evaluation method that divides the evaluation level into only "good and bad", and which is also different from the percentage system and the thousand-point system of the unlimited-level evaluation level. That is, reasonably dividing the evaluation level into five levels, the real feedback expert’s risk assessment for the urban large-scale public space is also conducive to the fusion of D-S evidence.

## 6. Conclusions

The main purpose of this work is to identify potential safety hazards in large-scale public spaces as early as possible, so that relevant management departments can promptly take measures to avoid risks. Taking Tian-yi Square in Ningbo as an example, this paper analyzes the scientific and feasible safety risk assessment index system from four aspects: environmental factors, human factors, infrastructure factors, and management factors. The risk assessment method using Dempster–Shafer theory is not limited to a single risk event, but also a comprehensive assessment method of the multiple risk factors faced by urban large-scale public spaces. The main results are described as follows.
(1)A robust risk indicator system for assessing the security risk of large-scale public spaces were selected by Delphi method and information entropy method, which included four first-level indicators and twenty second-level indicators. The first-level indicators were environmental factors, human factors, infrastructure factors, and management factors. There was five second-level indicators belonging to the environmental factors, which were architectural layout, weather, external traffic, public health, and social stability; and there was also five second-level indicators belonging to the human factors, which were crowd characteristics, negligence behavior, macroscopic fundamental diagram, safety awareness, and panic. Meanwhile, there was five second-level indicators belonging to the infrastructure factors, which were exit, guide sign, alarm system, firefighting system, broadcasting and monitoring system; and there was also five second-level indicators belonging to the management factors, which were personnel training, public health management, event organization and management, emergency evacuation management, and infrastructure management.(2)Data were collected in Ningbo, using an expert questionnaire survey approach based on the RAIS. The survey results showed that the risk indicator system for the ULPS assessment process is scientific and reasonable. In the risk index system, twelve variables were found to be statistically significant, which were ranked by weights: emergency evacuation management, personnel training, crowd characteristic, macroscopic fundamental diagram, public health management, exit, event organization and management, broadcasting monitoring system, public health environment, negligence behavior, weather, alarm system, infrastructure management, panic, external traffic, guide sign, safety awareness, firefighting system, architectural layout, and social stability. In addition, in the calculation process of weights, we fully considered the expert’s knowledge background and opinions, and reduce the uncertainty in the assessment, which was characterized by strong explanatory and high precision.(3)A Dempster-Shafer Theory with evidence fusion technique was employed to analyze the interaction between the RAIS and risk level in the ULPS. The results from the DST approach indicated that three variables were found to be excellent level, which were ranked by importance: environmental factors, human factors, infrastructure factors. Only one variable was found to be good level, which were management factors. The results from the value of the MASS function indicated that three indicators were found in higher risk level, which were guiding signs, alarm system, and personnel training. Simultaneously, eleven indicators were found in a higher safety level, which were weather, external traffic, public health environment, social stability, crowd characteristic, negligence behavior, macroscopic fundamental diagram, panic, exit, broadcasting monitoring system, public health management. The findings of this study provided insight into the factors associated with environmental factors, human factors, infrastructure factors, and management factors in the ULPS.

In the risk assessment of large-scale public spaces, we used the DST approach to conduct a multi-level risk assessment, and the risk grading of each evaluation index was also obtained. Taking Tian-yi Square as a case study, the results were consistent with the actual operation of large-scale public spaces, indicating that the DST approach has certain theoretical guiding significance and practical value. 

However, there are some limitations in this study. Firstly, the survey was only conducted in Ningbo city, but studies based on multiple cities could help better understand and capture more risk factors affecting large-scale public spaces. Secondly, an update questionnaire survey and risk assessment index system could be conducted to capture more meaningful factor with risk assessment work. We have to accept that there are other indicators in the risk assessment of large-scale public spaces, such as building characteristic index (e.g. building density, height, fire resistance, ventilation, etc.) in Environmental Factors (A1), temporary shelter, emergency shelter, basement shelter, and flame retardant equipment in Infrastructure Factors (A3), and evacuation training, safety education in Management Factors (A4). Hence, it will be encouraged to build a risk assessment indicators database for large-scale public spaces in the future study. Thirdly, risk management was defined as a procedure to control the level of risk and to mitigate its effect. Hence, the generally steps of risk management in the ULPS could be captured and described, such as risk identification, risk analysis, risk response and etc.

Furthermore, the extension of this work should examine the different expert weights and experience scores for the RAIS, especially for unobserved heterogeneity across more experts or staff or passengers or tourists. To solve this problem, in the future study, we will establish a historical database to quantify risk assessment indicators to reduce the impact of expert subjective factors on the assessment results. Recent work only provided the framework of discernment DST model [38,39,40,41,42,43,44]. Under this framework, risk assessment effects on environmental factors, human factors, infrastructure factors, and management factors across various passengers (or tourists) in the large-scale public spaces can be estimated, we are expecting more research results to emerge by standing upon the shoulders of ours. 

## Figures and Tables

**Figure 1 ijerph-16-02942-f001:**
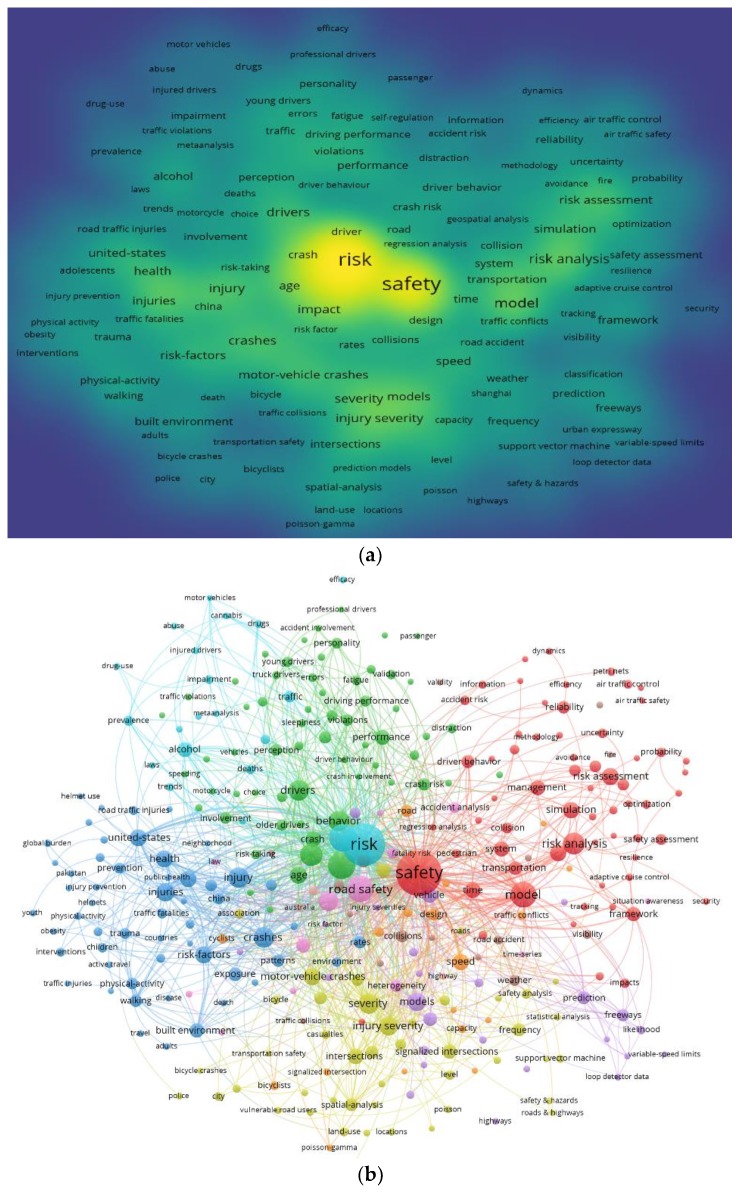
The keywords in risk assessment studies, 1997–2019. (**a**) The density of main research keywords; (**b**) the keywords co-occurrence network of risk analysis studies.

**Figure 2 ijerph-16-02942-f002:**
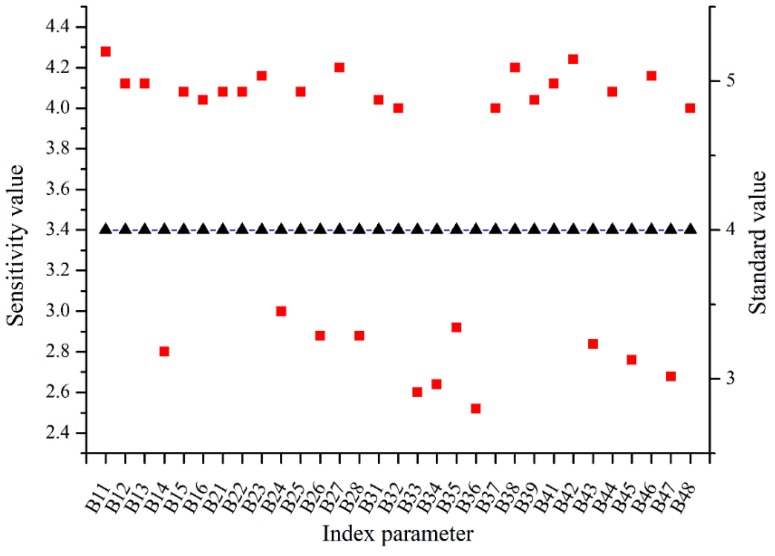
Result of sensitivity analysis of indicators.

**Figure 3 ijerph-16-02942-f003:**
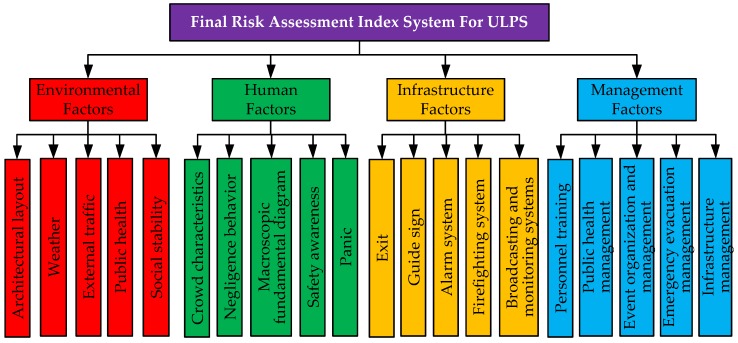
Result of the final risk assessment index system for Urban Large-scale Public Spaces (ULPS).

**Table 1 ijerph-16-02942-t001:** Statistical results of casualty incidents in the Urban Large-scale Public Spaces (ULPS) across the world (2000–2018) [1].

Time	Place	Type	Death	Injury	Reason
9 March 2000	Jiaozuo, China	Theatre	74	/	Fire
25 March 2000	Durban, South Africa	Ballroom	13	150	Fire
9 May 2001	Ghana	Stadium	100	/	Crowd
3 September 2001	Shaanxi, China	Yuquan Park	16	6	Crowd
20 February 2002	Egypt	Train	350	/	Fire
19 February 2003	South Korea	Train	120	387	Arson
5 February 2004	Beijing, China	Lantern Show	37	15	Crowd
11 March 2004	Spain	Station	200	1800	Terrorist Attack
1 September 2004	Russia	School	300	/	Terrorist Attack
10 June 2005	Shantou, China	Hotel	31	15	Fire
7 July 2005	London, UK	Metro	52	700	Terrorist Attack
12 September 2006	Yemen	/	51	200	Crowd
18 July 2007	Jinan, China	/	25	170	Rainstorm
3 August 2008	India	Temple	162	47	Crowd
5 July 2009	Urumqi, China	/	184	939	Terrorist Attack
15 November 2010	Shanghai, China	Apartment	85	71	Fire
15 January 2011	India	Temple	102	100	Crowd
18 March 2012	Egypt	Gathering	3	137	Crowd
28 October 2013	Beijing, China	Square	5	40	Terrorist Attack
1 March 2014	Kunming, China	Train Station	29	143	Terrorist Attack
13 November 2015	France	Stadium	197	/	Terrorist Attack
21 May 2016	Dalian, China	Market	3	/	Fire
25 May 2017	Henan, China	Senior Apartment	39	6	Fire
14 May 2018	Bengal	Ramadan Rally	10	50	Crowd
25 August 2018	Harbin, China	Hotel	20	23	Fire
9 September 2018	Luanda	Stadium	5	7	Crowd
17 March 2019	Northern Ireland.	Hotel	3	/	Human Crush
26 June 2019	Madagascar	Stadium	16	101	Human Crush

**Table 2 ijerph-16-02942-t002:** Preliminary risk assessment index system.

First-Level Indicators	Second-Level Indicators	Explanations
Environmental factors (A1)	Architectural layout (B11)	These indicators reflect the relationship between various environmental factors and safety risks in UPLS, and reflect the impact of random factors on safety risk assessment.
	Weather (B12)	
	External traffic (B13)	
	Internal traffic (B14)	
	Public health (B15)	
	Social stability (B16)	
Human factors (A2)	Crowd characteristics (B21)	These indicators reflect the relationship between human factors and risk level in UPLS, and reflect the impact of human factors on risk assessment level of UPLS.
	Negligence behavior (B22)	
	Macroscopic fundamental diagram (B23)	
	Age and gender (B24)	
	Safety awareness (B25)	
	Physical fitness (B26)	
	Panic (B27)	
	Trip purpose (B28)	
Infrastructure factors (A3)	Exit (B31)	These indicators reflect the relationship between infrastructure factors and risk level of UPLS, and reflect the impact of static factors on risk assessment level of UPLS.
	Guide sign (B32)	
	Power system (B33)	
	Lighting system (B34)	
	Fume exhaust system (B35)	
	Drainage system (B36)	
	Alarm system (B37)	
	Firefighting system (B38)	
	Broadcasting and monitoring system (B39)	
Management factors (A4)	Personnel training (B41)	These indicators reflect the relationship between the management mechanism of public places and risk level of public places, and reflect the impact of dynamic factors on risk assessment level of UPLS.
	Public health management (B42)	
	Security risk management (B43)	
	Event organization and management (B44)	
	Fire safety management (B45)	
	Emergency evacuation management (B46)	
	Store management (B47)	
	Infrastructure management (B48)	

**Table 3 ijerph-16-02942-t003:** Scores of Risk Assessment Index System (RAIS) by experts.

Indicators	The importance Value of RAIS ^a^					The level Value of RAIS ^b, d^				
	E1 ^c^	E2	E3	E4	E5	E1	E2	E3	E4	E5
Environmental factor	2	3	4	2	3	/	/	/	/	/
Human factor	2	2	2	3	1	/	/	/	/	/
Equipment factors	3	2	3	1	3	/	/	/	/	/
Management factors	1	1	1	1	2	/	/	/	/	/
Architectural layout	5	4	2	3	5	8	6	7	9	7
Weather	2	2	3	2	4	9	9	8	9	10
External traffic	3	3	4	4	3	7	8	9	8	9
Public health environment	1	1	1	3	2	9	10	8	8	9
Social stability	4	5	5	5	4	10	9	10	10	9
Crowd characteristic	3	3	2	3	1	7	9	8	8	10
Negligence behavior	4	1	4	3	4	9	8	9	8	8
Macroscopic fundamental diagram	1	2	2	4	2	9	8	9	8	9
Safety awareness	5	5	4	5	4	7	6	7	6	8
Panic	2	4	5	2	5	8	9	7	7	8
Exit	2	1	1	2	1	9	7	8	7	9
Guide sign	5	3	2	4	2	5	6	5	6	4
Alarm system	3	3	4	3	3	5	5	5	6	6
Firefighting system	4	4	4	5	4	6	6	7	7	6
Broadcasting monitoring system	1	2	2	1	2	10	10	9	9	10
Personnel training	1	2	2	1	2	6	5	7	6	8
Public health management	4	2	1	2	3	8	9	8	8	7
Infrastructure management	3	4	4	4	5	9	6	7	8	7
Event organization and management	2	3	3	3	4	8	7	7	6	8
Emergency evacuation management	2	1	2	1	1	8	6	6	8	7

Note: ^a^ denotes the importance value of risk assessment index system (RAIS), which is from 1–5. ^b^ denotes the level value of risk assessment index system (RAIS), which is from 1–10. ^c^ means the Expert 1. ^d^ a five-level risk assessment standard is proposed by the standard [47,48], the description of which is excellent, good, satisfactory, fair and poor. Next, the use of 10-scores scale that defines criterion for the hazard, is generally preferable to quantitative analysis, which are excellent (8 ~ 10], good (6 ~ 8], satisfactory (4 ~ 6], fair (2 ~ 4] and poor (0 ~ 2].

**Table 4 ijerph-16-02942-t004:** Results of the weights of indicators in Tian-yi Square.

First-Level Indicators	Weights	Second-Level Indicators	Weights
Environmental factor	0.200	Architectural layout	0.155
		Weather	0.231
		External traffic	0.208
		Public health environment	0.271
		Social stability	0.135
Human factor	0.259	Crowd characteristic	0.252
		Negligence behavior	0.201
		Macroscopic fundamental diagram	0.246
		Safety awareness	0.139
		Panic	0.161
Equipment factors	0.230	Exit	0.254
		Guide sign	0.160
		Alarm system	0.193
		Firefighting system	0.144
		Broadcasting monitoring system	0.249
Management factors	0.311	Personnel training	0.238
		Public health management	0.193
		Infrastructure management	0.140
		Event organization and management	0.186
		Emergency evacuation management	0.243

**Table 5 ijerph-16-02942-t005:** Results of Mass function of secondary indicators.

Secondary Indicators	R_1_	R_2_	R_3_	R_4_	R_5_	Θ
Architectural layout	0.002	0.998	0.000	0.000	0.000	0.000
Weather	1.000	0.000	0.000	0.000	0.000	0.000
External traffic	1.000	0.000	0.000	0.000	0.000	0.000
Public health environment	1.000	0.000	0.000	0.000	0.000	0.000
Social stability	1.000	0.000	0.000	0.000	0.000	0.000
Crowd characteristic	1.000	0.000	0.000	0.000	0.000	0.000
Negligence behavior	1.000	0.000	0.000	0.000	0.000	0.000
Macroscopic fundamental diagram	1.000	0.000	0.000	0.000	0.000	0.000
Safety awareness	0.000	0.999	0.001	0.000	0.000	0.000
Panic	1.000	0.000	0.000	0.000	0.000	0.000
Exit	1.000	0.000	0.000	0.000	0.000	0.000
Guide sign	0.000	0.000	1.000	0.000	0.000	0.000
Alarm system	0.000	0.000	1.000	0.000	0.000	0.000
Firefighting system	0.000	0.995	0.005	0.000	0.000	0.000
Broadcasting monitoring system	1.000	0.000	0.000	0.000	0.000	0.000
Personnel training	0.000	0.577	0.423	0.000	0.000	0.000
Public health management	1.000	0.000	0.000	0.000	0.000	0.000
Infrastructure management	0.002	0.998	0.000	0.000	0.000	0.000
Event organization and management	0.002	0.998	0.000	0.000	0.000	0.000
Emergency evacuation management	0.000	0.886	0.114	0.000	0.000	0.000

**Table 6 ijerph-16-02942-t006:** Results of Mass function of primary indicators.

First-Level Indicators	R_1_	R_2_	R_3_	R_4_	R_5_	Θ
Environmental factors	1.000	0.000	0.000	0.000	0.000	0.000
Human Factors	1.000	0.000	0.000	0.000	0.000	0.000
Infrastructure factors	0.000	0.000	1.000	0.000	0.000	0.000
Management factor	0.000	1.000	0.000	0.000	0.000	0.000

**Table 7 ijerph-16-02942-t007:** Results of Mass function of First-level indicators.

Primary Indicators	Excellent ^a,b^	Good	Satisfactory	Fair	Poor
Environmental factors	0.845	0.155	0.000	0.000	0.000
Human factors	0.861	0.139	0.000	0.000	0.000
Infrastructure factors	0.503	0.143	0.354	0.000	0.000
Management factors	0.194	0.678	0.128	0.000	0.000

Note: **a** The criterion for judging risk level come from Production Safety Law of the People’s Republic of China [47], Detailed rule for the management and control system of electricity enterperise work safety risk classification [47], and DoD standard which is approved for use by all Military Departments and Defense Agencies within the Department of Defense (DoD) [48]. **b** Following the standard [47,48], in our study, a five-level risk assessment standard is proposed, the description of which is excellent, good, satisfactory, fair and poor. Next, the use of 10-scores scale that defines criterion for the hazard, is generally preferable to quantitative analysis, which are excellent (8~10], good (6~8], satisfactory (4~6], fair (2~4] and poor (0~2].

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
