# Peer review of "Risk Assessment in Urban Large-Scale Public Spaces Using Dempster-Shafer Theory: An Empirical Study in Ningbo, China"

_ijerph, 2019, doi:10.3390/ijerph16162942_

Round 1

Reviewer 1 Report

This study introduced a risk assessment method based on Dempster–Shafer theory, and they used a case in Ningbo city to validate its effectiveness. The whole paper is easy to follow. My major comments include:

1. Line 330, authors stated that they used expert survey results to determine the weights. How to estimate the rationality of the evaluating results, I means whether the results would be various according to different experts in the questionnaires, how to avoid this sensibility?

2. Line 356, What is the “Normal Cloud Model”? the authors did not provide any information about it and how to use it to estimate values of parameters.

3. What is the main contribution of this study?

Author Response

Please refer to the attached pdf version

Reviewer 2 Report

Email Author

The risk assessment  research is very interesting

Introduction section shall highlight the need for the study.

Methodology is appropriate for the study. However, some of the statements required more clarifications.

Research limitations must be explained.

Implications could and should be better explored and developed. The implications for research, theory, practice and society are not clear though I can see that these aspects can be elaborated further. The authors need to conduct and familiarise with the academic literature surrounding the subject matter to answer the implications for research, practice and/or society much clearer.

The literature used in this paper is quite varied and up-to-date. However the authors should supplement it with:

- “Total Efficient Risk Priority Number (TERPN): a new method for risk assessment”

 G. Di Bona, A. Forcina, A Silvestri, A.Petrillo 
Journal of Risk Research Volume 21, Issue 11, 2 November 2018, Pages 1384-1408

Author Response

(The authors gave the same response as above.)

Reviewer 3 Report

Urban large-scale public spaces safety assessment has played a crucial role in the theory and practice of urban sustainable development. It is necessary to identify potential safety hazards in large-scale public spaces, so that relevant management departments would take measures to avoid risks promptly. This study is very interesting and valuable. Our team has always been concerned about the problem of the risk assessment in urban large-scale public spaces. There are some suggestions for your reference.

1In the part of literature review, there are some visualizations with a longer length, including the research countries, institutions, researchers and their cooperation. What is the role for the research content, research methods and research conclusions of this paper? In the end, it is concluded that there is a paucity of research discerning the interrelationships between the multiple-index system and risk assessment, and the contributory indicators and multiple-index weight at a disaggregated level using Delphi method and entropy method. Does the above reasons could be used as a basis for this study? In my opinion, we should focus on the problem of this study, and it would be better to make a problem-oriented literature review.

2What are the principles for selecting the preliminary risk assessment index system in this study (e.g., Table 1)? I think there should be give some explanations in this paper. In addition, is there any difference in risk assessment index system of different types of large public spaces for evaluating

3In Section 4.4, there are some descriptions about the outcome of risk assessment. What is the criterion for judging Execllent or Good? In addition, does located in the southeast coastal area and belongs to the subtropical monsoon climate could be a reason for the risk assessment level of Tian-yi Square is safe?

4In this study, is the result of risk assessment reasonable? How to prove the rationality of the research methods used in this paper?

5There are some problems in English language and style of the manuscript. For example, there are many spelling errors, including the same and repeated sentences, abbreviations with inconsistency and errors (e.g. Section 4). I suggest that the author conduct a thorough examination.

Author Response

(The authors gave the same response as above.)

Round 2

Reviewer 1 Report

In this revision, authors responded all my concerning comments, and I think the quality of the paper has improved largely. Before I address my acceptance of this study, I suggest authors could add several papers focusing on crash risk analysis and modeling in the introduction and reference, these papers could be helpful for their future researches.

[1] Tang, J. et al., Crash injury severity analysis using a two-layer Stacking framework”. Accident Analysis and Prevention, Vol.122, 226-238, 2019.

[2] Zou, Y. et al., A copula-based approach for accommodating the underreporting effect in wildlife-vehicle crash analysis. Sustainability. No.11, Vol.418, 1-13, 2019

[3] Zong, F., et al., Analyzing Traffic Crash Severity with Combination of Information Entropy and Bayesian Network”. IEEE Access. Vol.7, 63288-63302, 2019

Author Response

We would like to thank the reviewers for their constructive comments, which have significantly improved the quality of the paper. We have revised the manuscript according to your first-round constructive suggestions, where the corresponding changes to your comments have been modified in the revised version of the manuscript. The second-round responses to your insightful comments are attached as follows.

The related papers focusing on crash risk analysis and modeling are added on Literature Review Chapter, the order of the corresponding references has also been revised. Tang, J.; Liang, J.; Han, C.; Li, Z.; Huang, H. (2019). Crash injury severity analysis using a two-layer stacking framework. Accid. Anal. Prev. 2019, 122, 226-238. Zou, Y.; Zhong, X.; Tang, J.; Ye, X.; Wu, L.; Ijaz, M.; Wang, Y. A copula-based approach for accommodating the underreporting effect in wildlife‒vehicle crash analysis. Sustainability 2019, 11, 418.  Zong, F.; Chen, X.; Tang, J.; Yu, P.; Wu, T. Analyzing traffic crash severity with combination of information entropy and Bayesian network. IEEE Access 2019, 7, 63288-63302.

Reviewer 3 Report

From the revision, I think this paper has been improved in most aspects greatly. It can be found that the writing of this paper is clearer, the logic is more rigorous, the structure is more complete, and the use of language has been improved. In addition, I also found several problems in the review for the author's reference:

(1) In the literature review part, I suggest that you could review the research methods adopted in this paper, rather than focus on there are few applications of this method; it is suggested that you could properly review the application of this method in other fields, and then elaborate the rationality and feasibility of using this method in this paper.

(2) It is suggested that you could check whether the risk assessment indicators listed in table 2 of this paper are comprehensive and whether there are other indicators in the risk assessment of large urban public space.

(3) I also suggest that you could reconfirm the following writing: (a) In line 394 and 395 of the paper, RSIS is the abbreviation of which term? (b) In lines 555 and 556 of the paper, the expression about “the value of the excellent level is 0.503 and the value of the good level is 0.143. The risk assessment level of infrastructure factors is good.” Is it good or excellent?

Author Response

We would like to thank the reviewers for their constructive comments, which have significantly improved the quality of the paper. We have revised the manuscript according to your first-round constructive suggestions, where the corresponding changes to your comments have been modified in the revised version of the manuscript. The second-round responses to your insightful comments are attached as follows.

Reviewer 3

From the revision, I think this paper has been improved in most aspects greatly. It can be found that the writing of this paper is clearer, the logic is more rigorous, the structure is more complete, and the use of language has been improved. In addition, I also found several problems in the review for the author's reference:

In the literature review part, I suggest that you could review the research methods adopted in this paper, rather than focus on there are few applications of this method; it is suggested that you could properly review the application of this method in other fields, and then elaborate the rationality and feasibility of using this method in this paper. Many thanks for your comment. Following the reviewer’s suggestion, the literature review part was reorganized and rewritten on Line 86 to 145, page 3 to 4. It is suggested that you could check whether the risk assessment indicators listed in table 2 of this paper are comprehensive and whether there are other indicators in the risk assessment of large urban public space. This is a good question. We have to admit that there are other indicators in the risk assessment of large-scale public spaces, such as building characteristic index (e.g. building density, height, fire resistance, ventilation, etc.) in Environmental Factors (A1), temporary shelter, emergency shelter, basement shelter, and flame retardant equipment in Infrastructure Factors (A3), and evacuation training, safety education in Management Factors (A4). In fact, it is an interesting and challenging issue to select the risk assessment indicator system (RAIS). To fill this gap, we have divided the whole selection process for a risk assessment indicator system into two periods. For the first period, the four principles, such as scientific principle, comprehensive principle, accurate principle and operable principle, were obeyed to select the preliminary RAIS. At this stage, 4 first-level indicators and 31 second-level indicators were built for the preliminary RAIS. For the second period, an index sensitivity analysis method was applied to calculate and identify the preliminary RAIS, a Delphi method was also employed to analyse the sensitivity of evaluation indicators, and the final 4 first-level indicators and 20 second-level indicators are obtained in our study. Following the reviewer’s suggestion, in the Conclusions part of this study, we have made more explanations for the risk assessment indicators in the future study on Line 601 to 606, page 22 of 28. I also suggest that you could reconfirm the following writing: (a) In line 394 and 395 of the paper, RSIS is the abbreviation of which term? (b) In lines 555 and 556 of the paper, the expression about “the value of the excellent level is 0.503 and the value of the good level is 0.143. The risk assessment level of infrastructure factors is good.” Is it good or excellent? Thank you for your helpful comments. (a) it is a mistake, we have modified RSIS to RAIS. (b) it is still a mistake, the sentence has been revised as “The risk assessment level of infrastructure factors is also excellent.” on Line 504-505, Page 19 of 28.
